# Dual *Mkk4* and *Mkk7* Gene Deletion in Adult Mouse Causes an Impairment of Hippocampal Immature Granule Cells

**DOI:** 10.3390/ijms22179545

**Published:** 2021-09-02

**Authors:** Rubén Darío Castro-Torres, Jordi Olloquequi, Miren Etchetto, Pablo Caruana, Luke Steele, Kyra-Mae Leighton, Jesús Ureña, Carlos Beas-Zarate, Antoni Camins, Ester Verdaguer, Carme Auladell

**Affiliations:** 1Department of Cell Biology, Physiology and Immunology, Biology Faculty, Universitat de Barcelona, 08028 Barcelona, Spain; dario.castro@cucba.udg.mx (R.D.C.-T.); pcaruanas@gmail.com (P.C.); luketaylorsteele@gmail.com (L.S.); kyramael34@gmail.com (K.-M.L.); jurena@ub.edu (J.U.); everdaguer@ub.edu (E.V.); 2Laboratory of Neurobiotechnology CUCBA, Department of Cell and Molecular Biology, Universidad de Guadalajara, Jalisco 45200, Mexico; carlos.beas@academicos.udg.mx; 3Laboratory of Cellular and Molecular Pathology, Health Sciences Faculty, Biomedical Sciences Institute, Universidad Autónoma de Chile, Talca 3460000, Chile; jordiig82@gmail.com; 4Department of Pharmacology, Toxicology and Therapeutic Chemistry, Pharmacy and Food Sciences Faculty, Universitat de Barcelona, 08028 Barcelona, Spain; mirenettcheto@ub.edu (M.E.); camins@ub.edu (A.C.); 5Centre for Biomedical Research of Neurodegenerative Diseases (CIBERNED), Instituto de Salud Carlos III, 28029 Madrid, Spain; 6Institut de Neurociències, Universitat de Barcelona, 08035 Barcelona, Spain

**Keywords:** Cre-LoxP, MKK4, MKK7, pJNK, DCX, hippocampus

## Abstract

(1) Background: The c-Jun-NH2-terminal protein kinase (JNK) is a mitogen-activated protein kinase involved in regulating physiological processes in the central nervous system. However, the dual genetic deletion of *Mkk4* and *Mkk7* (upstream activators of JNK) in adult mice is not reported. The aim of this study was to induce the genetic deletion of *Mkk4/Mkk7* in adult mice and analyze their effect in hippocampal neurogenesis. (2) Methods: To achieve this goal, *Actin*-*Cre*^ERT2^ (*Cre*^+/^^−^), *Mkk4**^flox/flox^*, *Mkk7**^flox/flox^* mice were created. The administration of tamoxifen in these 2-month-old mice induced the gene deletion (*Actin*-*Cre*^ERT2^ (*Cre*^+/−^), *Mkk4^∆/∆^*, *Mkk7^∆/∆^* genotype), which was verified by PCR, Western blot, and immunohistochemistry techniques. (3) Results: The levels of MKK4/MKK7 at 7 and 14 days after tamoxifen administration were not eliminated totally in CNS, unlike what happens in the liver and heart. These data could be correlated with the high levels of these proteins in CNS. In the hippocampus, the deletion of *Mkk4/Mkk7* induced a misalignment position of immature hippocampal neurons together with alterations in their dendritic architecture pattern and maturation process jointly to the diminution of JNK phosphorylation. (4) Conclusion: All these data supported that the MKK4/MKK7–JNK pathway has a role in adult neurogenic activity.

## 1. Introduction

The c-Jun NH2-terminal kinases (JNKs) are members of the Mitogen-Activated Protein Kinases (MAPKs) super-family. JNK activity regulates several cellular functions, such as cell growth, differentiation, survival, and apoptosis through their impact on gene expression, cytoskeletal protein dynamics, and cell death/survival pathways. In fact, the best-described mechanism linked to the JNK pathway signaling is its pro-apoptotic action following sustained or intense exposure to cellular stress (including oxidative, genotoxic, and osmotic stress) or pro-inflammatory cytokines such as tumor necrosis factor (TNF)-α and interleukin (IL)-1β. This allows regulating physiological and pathological processes in autoimmune diseases, diabetes, cancer, cardiac hypertrophy, and neurodegenerative diseases [1,2,3,4,5].

Two MAP Kinase Kinases (MAPKKs)─MKK4 and MKK7─activate JNK proteins through dual phosphorylation at threonine and tyrosine residues (TPY-motif). In turn, MKK4 and MKK7 are activated by multiple MAP Kinase Kinase Kinase (MAPKKK) and MAPKKKKs, including the Mixed-Lineage Kinase (MLK), Transforming Growth Factor β-Activated Kinase (TAK), Apoptosis Signal-regulating Kinase (ASK), MAPK/RK Kinase Kinase (MEKKs), Dual Leucine Zipper Kinase (DLK), and their different isoforms in response to different stimuli. The scaffold JNK-Interacting Proteins (JIP) facilitate this sequential phosphorylation cascade. Their inactivation is regulated by Mitogen-Activated Protein Kinase Phosphatase (MKP)-1 and-7 [6].

MKK4 and MKK7 are crucial in the central and peripheral nervous systems’ (CNS and PNS) developmental processes, such as commissural fibers development, cell migration, and correct positioning of neuronal cells [7,8]. Indeed, it has been evidenced that MKK4 has dynamic changes during embryogenesis and postnatal development not only in the brain but also in other organs, such as liver and thymus [9]. In this way, *Mkk4* knock-out (KO) mice display altered hepathogenesis and die early in embryonic development, specifically between days 10.5 and 12.5, as also occurs to the *c-jun*^−/−^ mice. Moreover, *Mkk4* has been identified as a tumor-suppressor gene [9]. In turn, *Mkk4* gene deletion has revealed a compensatory effect of *Mkk7* [3,10,11]. MKK4 and MKK7 proteins are differentially located in neurons: while MKK4 is present in the cell body and their processes (dendrites and axons), MKK7 is mainly detected in the nucleus. Therefore, whereas both MKK4 and MKK7 can stimulate JNK in the nuclear fraction, MKK4 activity is likely to take part in maintaining the high basal activity on neurites. Consequently, MKK4 seems to be the kinase that mediates JNK dendritic outgrowth and establishes neural circuits in the brain [7].

Studies using KO mice with a single genetic mutation or mutation combinations of JNK1, JNK2, and JNK3 isoforms reported valuable data about their function [12]. Accordingly, *jnk3*-null mice show a reduction of c-JUN phosphorylation in ischemia–hypoxia [13] and in excitotoxicity [14], while *Jnk1*^−/−^ mice display progressive degeneration of long nerve fibers together with alterations in microtubule stability, evidencing a role of JNK1 in axonal growth and dendritic architecture maintenance [15,16].

In addition, the use of *Jnk1*^−/−^, *Jnk2^−/^*^−^, and *Jnk3*^−/−^ mice supported the fact that the JNK signaling pathway controls adult neurogenesis [17,18]. However, the specific mechanisms by which the molecular effectors of the JNK pathway are involved in all these processes remain obscure. Moreover, KO conventional murine models have significant limitations since the deletion key members of the pathway such as *Mkk4* or *Mkk7* and the double deletion *Jnk1/Jnk2* affect embryonic development and induce lethality [7,8,19,20]. In this sense, the conditional KO mice for *Mkk4* and *Mkk7* created with the Cre-LoxP system and *Nestin* promoter surpass the embryonic lethality, but the animals die soon after birth [7,8].

To overcome these limitations, the present study aimed to assess the role of MKK4 and MKK7 in immature hippocampal neurons and dendritic architecture maintenance, using a new conditional-induced KO mice, the hemizygous *Actin-**Cre*^ERT2^ (*Cre*^+/−^), *Mkk4**^flox/flox^, Mkk7^flox/flox^*. These mice allow a controlled deletion of *Mkk4* and *Mkk7* genes in postnatal stages or adults.

The results obtained in the present work demonstrate that the *Mkk4^∆/∆^Mkk7^∆/∆^* genotype mice can be induced in adulthood, showing a decrease in MKK4 and MKK7 proteins in different areas of the CNS (hippocampus, cortex, and cerebellum) as well as in heart and liver. This protein depletion was correlated with a reduction of JNK phosphorylation. Moreover, the analyses of immature neurons in the subgranular zone (SGZ) of the hippocampus in these KO *Mkk4^∆/∆^Mkk7^∆/∆^* mice evidenced modifications in their distribution and dendritic pattern when compared with wild-type (*WT*) mice. Finally, alterations in the dendritic distribution pattern of cortical neurons were also detected.

## 2. Results

### 2.1. Characterization of Mkk4 and Mkk7 Gene Deletion and Protein Elimination in the Hippocampus of Actin-Cre^ERT2^ (Cre^+/−^), Mkk4 ^Δ^^/^^Δ^, Mkk7^Δ/^^Δ^ Adult Mice

The relative quantification of MKK4 and MKK7 proteins was evaluated in the CNS, heart, and liver in two-month-old WT mice by immunoblot assays. The results showed that the amounts of MKK4 and MKK7 were higher in the CNS than in heart and liver (Figure 1A–C). In addition, the lower protein abundance of MKK7 was found in the heart and liver (Figure 1A,C).

Since *Mkk4 and Mkk7* genetic disabling results in a lethal embryonic phenotype [7,19,21], we generated a conditional KO hemizygous *Actin*-*Cre*^ERT2^ (*Cre*^+/−^), *Mkk4^flox/flox^Mkk7^flox/flox^* mice by crossing different mice colonies. Double KO *Actin*-*Cre*^ERT2^ (*Cre*^+/−^), *Mkk4^∆/∆^*, *Mkk7^∆/∆^* mice were induced after tamoxifen administration at 2 months old. The most effective tamoxifen treatment to obtain the double deletion was a dose of 10 mg.

A deletion of *Mkk4* (Figure 2A) and *Mkk7* (Figure 2B) was detected by PCR tail DNA genotyping. This deletion was detected in *Actin*-*Cre*^ERT2^ (*Cre*^+/−^)*, Mkk4^∆/∆^, Mkk7^∆/∆^* mice after 3 (data not shown), 7, and 14 days of tamoxifen treatment (Figure 2A,B). On the contrary, the deletion was not observed in *Actin*-*Cre*^ERT2^ (*Cre*^−/−^)*, Mkk4^flox/flox^Mkk7^flox/flox^* mice treated with tamoxifen and vehicle (Figure 2A,B). However, in *Actin*-*Cre*^ERT2^ (*Cre*^+/−^) *Mkk4^flox/flox^Mkk7^flox/flox^* mice treated with vehicle, a band of 300 bp was observed, indicating low expression (Figure 2B). This was to be expected, as there is evidence that some inducible *Cre* mice lines may undergo spontaneous recombination [22]. However, *Actin*-*Cre*^ERT2^ (*Cre*^+/−^), *Mkk4^flox/flox^Mkk7^flox/flox^* showed reduced spontaneous recombination activity and does not alter MKK4 and MKK7 protein expression. In addition, *Actin*-*Cre*^ERT2^ (*Cre*^−^^/−^)*, Mkk4^flox/flox^Mkk7^flox/flox^* and WT mice were used as a negative control for the subsequent experiment.

The hippocampus was used to elucidate the time necessary to diminish MKK4 and MKK7 protein levels. As shown by immunoblot, this decline was evident after 7 days of tamoxifen administration, and it was more noticed after 14 days (Figure 2C). This result was supported by the immunohistochemistries against MKK4 and MKK7 in the hippocampus of *Actin*-*Cre*^ERT2^ (*Cre*^+/−^), *Mkk4^∆/∆^*, *Mkk7^∆/∆^* mice after 14 days of tamoxifen administration. Immunohistochemistries showed that MKK4 and MKK7 decline their expression in *Actin*-*Cre*^ERT2^ (*Cre*^+/−^), *Mkk4^∆/∆^*, *Mkk7^∆/∆^* (Figure 2D).

It is important to note that not complete removal of the proteins was achieved from both MKK4 and MKK7 in the hippocampus; for this reason, this experimental model resembles a knock-down. However, the animals presented phenotypic alterations after 16 days of tamoxifen administration, such as signs of distress, weight loss, and reduced activity. Since mortality was high after 20 days of tamoxifen administration, all experiments were carried out 14 days after tamoxifen administration. At this time, we achieved the maximum levels of MKK4 and MKK7 protein reduction without any phenotype alteration.

The elimination of MKK4 and MKK7 proteins was screened in the liver, heart, and CNS of Actin-Cre^ERT2^ (Cre^+/^*^−^*), Mkk4^∆/∆^, Mkk7^∆/∆^ mice and their control mice groups (Table 1). The analyses were done through Western blot, using antibodies against MKK4 (Figure 3) and MKK7 (Figure 4). We evaluated protein levels in WT and the other experimental groups at 14 days after vehicle or tamoxifen administration. A reduction in MKK4 and MKK7 levels was only detected in Actin-Cre^ERT2^ (Cre^+/^*^−^*), Mkk4^∆/∆^, Mkk7^∆/∆^ mice (CRE (+) + T group). There was a higher reduction or apparent absence of these proteins in the heart and liver. The decreases in MKK4 levels achieved were ≈80% in cortex, ≈60% in hippocampus, ≈52.4% in cerebellum, ≈83.2% in liver, and ≈98% in heart (Figure 3). Meanwhile, the expression levels of MKK7 dropped ≈66% in the hippocampus, ≈56% in cortex, and ≈60% in cerebellum of Actin-Cre^ERT2^ (Cre^+/^*^−^*), Mkk4^∆/∆^, Mkk7^∆/∆^ mice (Figure 4). MKK7 disappeared in heart and liver after 14 days of tamoxifen administration (data not shown).

### 2.2. Genetic Ablation of Mkk4 and Mkk7 Reduces Phosphorylation of JNK in the Hippocampus 

The phosphorylation of JNK (pJNK) in the hippocampus was evaluated in WT, Actin-Cre^ERT2^(Cre^−/−^), Mkk4^flox/flox^, Mkk7^flox/flox^, and Actin-Cre^ERT2^(Cre^+/−^), Mkk4^∆/∆^, Mkk7^∆/∆^ mice in a time course of 3, 7, and 14 days after tamoxifen administration. Basal levels of JNK phosphorylation from Actin-Cre^ERT2^(Cre^−/−^), Mkk4^flox/flox^, Mkk7^flox/flox^ mice were higher than those of WT mice (Figure 5A). A significant reduction of JNK phosphorylation after 7 and 14 days of tamoxifen was observed in Actin-Cre^ERT2^(Cre^+/−^), Mkk4^∆/∆^, Mkk7^∆/∆^ mice when compared with WT and Actin-Cre^ERT2^(Cre^−/−^), Mkk4^flox/flox^, Mkk7^flox/flox^ mice (Figure 5A,B).

### 2.3. Immature Hippocampal Neurons in Actin-Cre^ERT2^(Cre^+/−^), Mkk4^∆/∆^, Mkk7^∆/∆^ Mice Were Misaligned and Showed Alterations in the Dendritic Pattern 

The immature neurons located in the SGZ of the hippocampus were analyzed in *Actin*-*Cre*^ERT2^(*Cre*^+/−^), *Mkk4^∆/∆^, Mkk7^∆/∆^* mice and compared with the ones in WT and *Actin*-*Cre*^ERT2^(*Cre*^−/−^), *Mkk4^flox/flox^, Mkk7^flox/flox^* mice. The analysis was performed by immunofluorescence against two well-known cell markers of immature neurons: Doublecortin (DCX) and Calretinin (CR), 14 days after tamoxifen administration. 

The quantification of immature neurons (DCX^+^ cells) revealed a slight decrease in those cells in *Actin*-*Cre*^ERT2^(*Cre*^−/−^), *Mkk4^flox/flox^, Mkk7^flox/flox^*, and *Actin*-*Cre*^ERT2^(*Cre*^+/−^), *Mkk4^∆/∆^, Mkk7^∆/∆^* mice compared to WT (Figure 6A,B). No differences were found in their number between *Actin*-*Cre*^ERT2^(*Cre*^−/−^), *Mkk4^flox/flox^, Mkk7^flox/flox^*, and *Actin*-*Cre*^ERT2^(*Cre*^+/−^), *Mkk4^∆/∆^, Mkk7^∆/∆^* mice (Figure 6B). However, DCX^+^ cells were mislocalized over the SGZ and Granular Cell Layer (GCL) in *Actin*-*Cre*^ERT2^(*Cre*^+/−^), *Mkk4^∆/∆^, Mkk7^∆/∆^* mice versus the other genotypes (Figure 6A). To evaluate this differential position between genotypes, we segmented manually the SGZ and GCL in three linear bins over the images obtained from immunofluorescence against DCX (Figure 6A). We defined them as sub-layer 1 (SGZ or GCL proximal to the hilus), sub-layer 2 (middle section of GCL), and sub-layer 3 (GCL proximal to molecular layer). The number of DCX^+^ cells in sub-layer 1 was lower in *Actin*-*Cre*^ERT2^(*Cre*^−/−^), *Mkk4^flox/flox^, Mkk7^flox/flox^* mice and *Actin*-*Cre*^ERT2^(*Cre*^+/−^), *Mkk4^∆/∆^, Mkk7^∆/∆^* mice compared to WT (Figure 6A). DCX^+^ cells in the sub-layers 2 and 3 were higher in *Actin*-*Cre*^ERT2^(*Cre*^+/−^), *Mkk4^∆/∆^, Mkk7^∆/∆^* mice than WT and *Actin*-*Cre*^ERT2^(*Cre*^−/−^), *Mkk4^flox/flox^, Mkk7^flox/^*^flox^ mice. Finally, the dendritic morphology of DCX^+^ cells was altered in *Actin*-*Cre*^ERT2^(*Cre*^+/−^), *Mkk4^∆/∆^, Mkk7^∆/∆^* mice, showing a loss in dendritic branches and a delay in arborization complexity (Figure 6C).

### 2.4. Reduction of MKK4 and MKK7 Proteins Alters the Late Differentiation of Immature Hippocampal Neurons 

An immunofluorescence against calretinin (CR), as a marker of early mature neurons, was performed after 14 days of tamoxifen administration. The results revealed that there was a reduction of the number of CR^+^ neurons in Actin-Cre^ERT2^(Cre^−/−^), Mkk4^flox/flox^, Mkk7^flox/flox^, and Actin-Cre^ERT2^(Cre^+/−^), Mkk4^∆/∆^, Mkk7^∆/∆^ mice compared with WT mice (Figure 7A). The difference was heightened in Actin-Cre^ERT2^(Cre^+/−^), Mkk4^∆/∆^, Mkk7^∆/∆^ mice (Figure 7B). In addition, the double immunofluorescence against DCX and CR showed that the number of DCX^+^/CR^+^ cells was equal in WT and Actin-Cre^ERT2^(Cre^−/−^), Mkk4^flox/flox^, Mkk7^flox/flox^ mice (Figure 8A), although the number of early mature cells (CR^+^) was lower in Actin-Cre^ERT2^(Cre^−/−^), Mkk4^flox/flox^, Mkk7^flox/flox^ mice (Figure 7A). However, the number of double-labeled cells (DCX^+^/CR^+^) was decreased in Actin-Cre^ERT2^(Cre^+/−^), Mkk4^∆/∆^, Mkk7^∆/∆^ mice compared with the other genotypes (Figure 8B).

### 2.5. Neuronal Dendritic Pattern Is Disorganized in Actin-Cre^ERT2^(Cre^+/−^), Mkk4^∆/∆^, Mkk7^∆/∆^ Mice 

Since the JNK pathway controls the stabilization of dendritic projections [15,16], immunofluorescence against microtubule-associated protein (MAP2) was used to analyze the dendritic pattern of neurons in the sensorimotor cortex. A disorganization of the dendritic field was observed in *Actin-Cre*^ERT*2*^(*Cre*^+/−^), *Mkk4^∆/∆^, Mkk7^∆/∆^* mice compared to WT and *Actin*-*Cre*^ERT2^(*Cre^−/−^),*
*Mkk4^flox/flox^, Mkk7^flox/flox^* mice (Figure 9). This finding supports that the decrease in JNK activity through MKK4 and MKK7 deletion interferes in neuronal architecture processes. Furthermore, the Hoechst stain revealed a loss of neural cells in upper cortical layers (II–III) in *Actin*-*Cre*^ERT2^(Cre^+/−^), *Mkk4^∆/∆^, Mkk7^∆/∆^* mice compared to WT and *Actin*-*Cre*^ERT2^(*Cre*^−/−^), *Mkk4*^flox/flox^, *Mkk7*^flox/flox^ mice (Figure 9).

## 3. Discussion

In the present work, we generated a new adult murine model with a double deletion (Mkk4^∆/∆^, Mkk7^∆/∆^ genotype). After 14 days of tamoxifen administration, Actin-Cre^ERT2^(Cre^+/−^), Mkk4^∆/∆^, Mkk7^∆/∆^ adult mice showed a reduction of MKK4 and MKK7 proteins in the CNS and other tissues, such as the heart and liver. This diminution provoked a decrease in JNK phosphorylation, which correlated with alterations in the position, dendritic pattern, and differentiation of immature hippocampal neurons as well as with changes in the dendritic pattern of cortical neurons. 

### 3.1. The Levels of MKK4 and MKK7 Are Not Equal in the Different Tissues

Actin-Cre^ERT2^(Cre^+/−^), Mkk4^∆/∆^, Mkk7^∆/∆^ mice overcome the developmental drawbacks of conventional KOs for Mkk4 or Mkk7 [23] and even some conditional KO mice, allowing MKK4 and MKK7 protein reductions in adults. Although the decrease was induced in all the body, it was not equal in all tissues, which was probably because of differences in their physiological basal levels. Thus, while protein elimination was apparently total in the heart and liver, it was partial in the CNS, supporting that MKK4 and MKK7 levels are higher in the CNS than in other tissues, which is in accordance with the results obtained with Western blot in WT mice (Figure 1). In this line, Lee et al. showed that the levels of Mkk4 transcripts were high in the cerebral cortex, hypothalamus, hippocampus, and cerebellum of adult mice [24]. All these data emphasized that the MKK4/MKK7/JNK signaling pathway has an important role in the adult CNS [25]. Moreover, the different subcellular localizations of these proteins supports that they have distinct functions, and therefore, their levels vary in distinct tissues [21,23]. This is supported by Tournier et al., who found that the simultaneous disruption of the Mkk4 and Mkk7 genes was required to block JNK activation caused by the exposure of cells to environmental stress (e.g., ultraviolet radiation) [26]. However, with stimuli such as pro-inflammatory cytokines (e.g., TNF and IL-1), the disruption of the Mkk7 gene alone could prevent JNK activation.

### 3.2. MKK7 Plays an Essential Role in Heart and Liver Tissues

After analyzing the levels of MKK4 and MKK7 in different tissues of Actin-Cre^ERT2^(Cre^+/−^), Mkk4^∆/∆^, Mkk7^∆/∆^ adult mice, we observed that the presence of MKK7 was reduced or absent in the heart and liver. These data are in accordance with those of Nishina et al., who evidenced low levels of MKK7 in embryonic tissue of mice, which was probably restricted to the skin, lung epithelium, and epithelial layers lining the olfactory cavity developing teeth. By contrast, MKK4 was ubiquitous and with high basal levels [21]. Despite the scarce levels of MKK7 detected in heart and liver, several studies support the notion that this protein has a critical role in these tissues, both in embryonic development and adulthood [7,27]. Indeed, Ooshio et al., through hepatocyte and hematopoietic cell-specific deletion of Mkk7, using Albumin (Alb)-Cre and Myxovirus resistance protein-1 (Mx1)-Cre line, evidenced that MKK7 is essential for wound-healing processes following parenchymal destruction by carbon tetrachloride (CCl_4_) in the liver [27]. In addition, MKK7 suppress branching morphogenesis through the modulation of hepatocyte-extracellular matrix interaction. 

Concerning MKK7 and cardiac tissue, Liu et al. revealed an essential protective role of this protein in the heart from hypertrophic insults in cardiomyocytes, hence preventing the transition to heart failure [28].

### 3.3. The Levels of JNK Phosphorylation Were Decreased in Actin-Cre^ERT2^(Cre^+/−^), Mkk4^∆/∆^, Mkk7^∆/∆^ Mice

MAP kinase cascade, which senses cellular and extracellular stress, conveys cellular response to regulate cell fate. The timing and duration of JNK activation determines whether cells proliferate or adapt to metabolic or toxic stress or undergo programmed cell death instead, such as apoptosis, necrosis, and even other forms of cell death. MKK4/MKK7 proteins have a role in the control of JNK activation by interacting with JNK via D-motif, phosphorylating JNK [29]. Since the levels of pJNK are correlated with the activity of this signaling pathway, and they are reduced with the Mkk4 and Mkk7 deletion, we circumscribed the analysis when Actin-Cre^ERT2^(Cre^+/−^), Mkk4^∆/∆^, Mkk7^∆/∆^ mice had a significant reduction of JNK phosphorylation at 14 days after tamoxifen administration. These data reinforce that Mkk4 and Mkk7 gene deletion correlated with JNK activity diminution.

### 3.4. The Deletion of Mkk4 and Mkk7 Gene Alters Immature Hippocampal Neurons 

Tangential-to-radial migration has been described for immature hippocampal neurons [30,31]. First, neuroblasts migrate tangentially after the last division from neuron stem cell clusters through the SGZ, and then apical dendrites extend toward the molecular layer [32]. In agreement with this, neuroblasts are lined up in the SGZ and have their apical dendrites projected in the radial direction both in WT and Actin-Cre^ERT2^(Cre^−/−^), Mkk4^flox/flox^, Mkk7^flox/flox^ mice. However, in Actin-Cre^ERT2^(Cre^+/−^), Mkk4^∆/∆^, Mkk7^∆/∆^ mice, these cells were disarranged, since they were displaced over the GCL, maintaining immature cell markers. These finding suggest that MKK4 and MKK7 have a role in adult neuroblast migration and differentiation processes, in accordance with the results obtained by Smith, Coker, and Tucker, who identified that the JNK signaling pathway is a regulator of branching and nucleokinesis during the migration of cortical interneurons [33]. In this respect, Nestin-Cre, Mkk7^flox/flox^ mice showed severe defects along embryonic brain development in radial migration and axonal growth [8]. In addition, Nestin-Cre, Mkk4^flox/flox^ mice showed misalignment of cerebellar granule cells and defects in radial migration [34]. However, no changes in cell differentiation were identified in Mkk4^flox/flox^ or Mkk7^flox/flox^ mice under Nestin-Cre promoter in developing ages, as it occurs in our adult mice, which is probably due to the combinatorial effect of the double deletion. Moreover, the analyses with Nestin-Cre mice have the disadvantage that even though they survive after birth, eventually, they die at postnatal day 21.

The alterations detected in the dendritic projections of immature hippocampal neurons and mature cortical neurons of Mkk4^∆/∆^ and Mkk7^∆/∆^ mice support that the MKK4/MKK7–JNK signaling pathway has a role in the maintenance of the dendritic and axonal processes [35] in accordance with Bjorkblom et al., who found that JNK phosphorylation of MAP2 plays an important role in defining dendritic architecture in the brain [36].

In this line, different studies reported that JNK1 regulates neural architecture through the phosphorylation of cytoskeletal substrates [36,37,38]. Further studies should be done to determinate how MKK4/MKK7/JNK signaling is involved in all these neuronal processes. 

To know the specific functions of the JNK signaling pathway in neural cell subpopulations, conditional KOs mice would be used with recombination under specific neural promoters instead of using the ubiquitous promoter Actin. In this way, there are CamKIIα-Cre mice that express Cre recombinase in postmitotic glutamatergic neurons of the CA1 hippocampus and layer V cerebral cortex [39,40] or Synapsyn I-Cre mice that drive the expression of Cre in general mature neurons [41]. Other Cre mice, such as Glial Fibrillary Acid Protein (GFAP)-Cre or GFAP-Cre^ERT2^, will allow the recombination glial linage [42,43] involved in the homeostatic functions control in health and disease.

## 4. Material and Methods

### 4.1. Animals 

Mice carrying homozygous floxed Mkk4 and Mkk7 genes were generated in our lab by crossing Mkk4^flox/flox^ mice [7] with Mkk7^flox/flox^ mice [44]. Double homozygous floxed mice (Mkk4^flox/flox^ Mkk7^flox/flox^ genotype) were selected and mated with conditional Actin-Cre^ERT2^ (Cre^+/−^) mice (see Appendix A). Thus, hemizygous (1) Actin-Cre^ERT2^(Cre^+/−^), Mkk4^flox/flox^, Mkk7^flox/flox^, (2) Actin-Cre^ERT2^(Cre^−/−^), Mkk4^flox/flox^, Mkk7^flox/flox^, and (3) C57BL/6 (WT mice) were used in this study. All mice were housed in constant and controlled environments during the experiments with a light/dark cycle of 12 h. The mice had free access to food and water. The experiments were conducted in accordance with the Council of Europe Directive 2010/63. The procedure was registered and accepted by the Catalan Government Decree 214/97, 30 July 2020, the University of Barcelona, and the Animal Experimentation Ethics Committee.

### 4.2. Tamoxifen Treatment

Actin-Cre^ERT2^(Cre^+/−^), Mkk4^flox/flox^, Mkk7^flox/flox^ 2-month-old mice were used to obtain a double KO mouse Mkk4 and Mkk7 (Actin-Cre^ERT2^(Cre^+/−^), Mkk4^∆/∆^Mkk7^∆/∆^ genotype). These mice have a Cre recombinase expressed under the Actin promoter and fused with human estrogen receptor that can be activated with tamoxifen as a selective estrogen receptor modulator. The CRE activation with tamoxifen allows controlling the specific time to delete floxed genes [45]. Tamoxifen was administered via orogastric gauge, at different doses and days, in order to evaluate the dose and time necessary to delete MKK4 and MKK7 proteins. The optimal dose found was 5 mg per day for two consecutive days. Tamoxifen (Sigma-Aldrich, Madrid, Spain) was dissolved in a solution containing 90% of sunflower oil and 10% ethanol. Actin-Cre^ERT2^(Cre^−/−^), Mkk4^flox/flox^, Mkk7^flox/flox^ mice were used as controls and were treated with vehicle solution (90% of sunflower oil/10% ethanol) or tamoxifen for two consecutive days. After 3 days, the Actin-Cre^ERT2^(Cre^+/−^), Mkk4^∆/∆^Mkk7^∆/∆^ genotype was detected. The protein elimination was screened in the heart, liver, and CNS at 3, 7, and 14 days after tamoxifen administration. The studies were done after 14 days of tamoxifen administration because beyond 16 days, the deterioration of treated animals was severe, since they showed signs of distress, weight loss, reduced activity, and after 20 days of treatment, there was high mortality (50%). Animals of each genotype used are shown in Table 1. Tamoxifen treatment is shown in Scheme 1.

### 4.3. Genotype Determination in Mice and Deletion on Tissue 

PCRs on tail DNA were used to identify offspring carrying the Mkk4^flox^ allele using forward (5′-GACATTGAGTTCCTTGCG-3′) and reverse (5′-TCCTATGTAGTAGGAGTTTG-3′) primers. Mkk4^+^ and Mkk4^flox^ alleles were identified with fragments of ≈390 bp and ≈490 bp, respectively. To find Mkk7^flox^ alleles, PCRs on tail DNA were performed using forward (5′-CTGCCTGTAGCATGCCCGAGCTGTC-3′) and reverse (5′-AGCTGTCTCATCTGTGCACCTCCCAGC-3′) primers, which gave fragments ≈290 bp for Mkk7^+^ and ≈390 bp for Mkk7^flox^ alleles. For detecting deletion after tamoxifen administration, PCRs on tail DNA and on brain tissue were performed using forward (5′-GGCAGCTTGTCAGATG-3′) and reverse (5′-TCCTATGTAGTAGGAGTTTG-3′) primers yielding ≈850 bp fragment for Mkk4^+^, ≈900 bp for Mkk4^flox^, and ≈450 bp for Mkk4^∆^. In addition, forward (5′-ATGCAGGCCATTGGGAAGTACCAAG-3′) and reverse (5′- AGAAAAATGAAGCCCGACTGTGCCT-3′) primers were used to identify Mkk7^∆^ alleles; since this PCR yielded only one band (300 bp for Mkk7^∆^), forward (5′-TGAGCGAGCTCATCAAGATAATCAGGT-3′) and reverse (5′-GTTAGCATTGAGCTGCAAGCGCCGTCT-3′) primers were also added to amplify the 550 bp fragment from the intron of the LC3 genome as internal control. We identified the Transgene Cre sequence using forward (5′-GCATTACCGGTCGATGCAACGAGTGATGAG-3′) and reverse (5′-GAGTGAACGAACCTGGTCGAAATCAGTGCG-3′) primers, yielding a 400 bp fragment. Finally, forward (5′-TGGACAGGACTGGACCTCTGCTTTCCTAGA-3′) and reverse (5′-TAGAGCTTTGCCACATCACAGGTCATTCAG-3′) primers to Intestinal Fatty Acid-Binding Protein (I-FABP) gene (200 bp fragment) were used as internal control. See Appendix A.

### 4.4. Preparation of Lysates 

From the CNS, we dissected the cortex, hippocampus, and cerebellum. In addition, liver and heart were used. Tissues were homogenized with lysis buffer (137 mM NaCl, 20 mM Tris-HCl, pH 8.0, 1% NP 40, 10% glycerol, 1 mM PMSF, 10 µg/mL aprotinin, 1 µg/mL leupeptin, and 0.5 mM sodium orthovanadate). Homogenates were spun at 13,000 rpm for 20 min at 4 °C, and the protein content of the supernatants was determined by the BCA method (Pierce Company, Rockford, MI, USA). A range of 20–50 µg of protein was mixed with a loading buffer (β-mercaptoethanol 100 mM, Tris-HCl pH 6.8, 2% Sodium Dodecyl Sulfate, SDS) and was denatured at 95 °C for 5 min.

### 4.5. Immunoblot Analysis 

Protein extracts were loaded in 12% SDS-PAGE (Sodium Dodecyl Sulfate-Polyacrylamide Gel Electrophoresis) at 90 V for 2–3 h and transferred overnight at 4 °C and 45 V to a PVDF membrane (0.45 µm, Millipore, Bedford, MA, USA). The membrane was blocked in 10% non-fat milk in TBS-Tween, pH 7.4, for 4 h at RT. Afterwards, the membrane was incubated with specific primary antibodies for MKK4 (1:1000, 9152S, Cell Signaling Technology, Leiden, The Netherlands), MKK7 (1:5000, ab52618, Abcam; and 1:1000, 4172, Cell Signaling Technology, USA), total JNK (1:1000, 9152S, Cell Signaling Technology, Leiden, The Netherlands), phospho-JNK (1:500, 9251S, Cell Signaling Technology, Leiden, The Netherlands), and GAPDH (1:20,000, 2118, Cell Signaling Technology, USA) O/N at 4 °C. After several washes, the membrane was further incubated with a HRP-linked secondary antibody, Anti-rabbit IgG, or anti-Mouse IgG (7074, 7076, Cell Signaling Technology, Leiden, The Netherlands) diluted at 1:2000 in TBS-Tween for 1 h at RT. The signals were developed with chemiluminescent substrate (ECL^TM^ Western Blotting Analysis System, GE Healthcare, Madrid, Spain) before film exposure (Medical X-ray film, Fujifilm (Rosex, Barcelona, Spain). GAPDH was used to normalize differences in gel loading. Semi-quantitative values were obtained using Image Lab software (Bio-Rad, Madrid, Spain). The size of the bands was determined using molecular weight markers (1610374, Bio-Rad).

### 4.6. Immunofluorescences 

Free-floating technique immunofluorescences were conducted in coronal sections of 20 µm. Animals were perfused with 40 g/L of paraformaldehyde in 0.1 mol/L of phosphate buffer. The brains were removed, subsequently rinsed in the same solution with 300 g/L of sucrose for 48 h, and frozen. Then, they were cut in a cryostat (Leica Microsystems, Wetzlar, Germany). Free-floating coronal sections were rinsed in 0.1 mol/L phosphate buffer (PB), pH 7.2. After that, brain slices were pre-incubated in a blocking solution (100 mL/L of fetal bovine serum (FBS) and 2% gelatin in PBS with 5 mL/L Triton X-100) at room temperature (RT). Then, the samples were incubated overnight (O/N) at 4 °C with different primary antibodies: goat anti-DCX (1:200, sc-8066, Santa Cruz Biotechnology, Heidelberg, Germany), mouse anti-MAP2 (1:1000, 015M4775V, Sigma), and rabbit anti-Calretinin (1:2000, 7699/4, Swant Inc, Burgdorf, Switzerland). The secondary antibodies used were Alexa Fluor 488 donkey anti-goat (1:200, A11055, Life technologies, Madrid, Spain), Alexa Fluor 594 goat anti-mouse (1:200, A11005, Thermo Fisher Scientific, Madrid, Spain), and Alexa Fluor 594 goat anti-rabbit (1:200, Thermo Fisher Scientific, A11012, Madrid, Spain). Sections were counter-stained using 0.1 µg/mL Hoechst 33,258 (Sigma-Aldrich, USA) for nuclear staining. Sections corresponding to the hippocampal levels between Bregma −1.28 and −2.12 mm, according to the Atlas reported by Paxinos and Watson [46] were used to analyze the hippocampus and cortex (3 animals/genotype, 4–8 sections/animal). 

### 4.7. Data Analysis 

Student’s *t*-test was performed to compare two conditions, and one-way ANOVA post hoc Fisher’s Least Significant Difference (LSD) tests were used for comparison in 3 or more conditions. Level of significance was fixed at α = 0.05. Both statistical analyses and graphs were created with the Graph Pad InStat software V5.0 (Graph Pad Software Inc., San Diego, CA, USA).

## 5. Conclusions

The new transgenic Actin-Cre^ERT2^ (Cre^+/−^), Mkk4^flox/flox^, Mkk7^flox/flox^ mice allow inducing the conditional deletion of Mkk4 and Mkk7 genes in adults, hence overcoming the lethality induced with other KOs. Thus, these types of mice would allow studying the specific functions of MKK4 and MKK7 proteins in adult organisms. Specifically, here, we identified the role that the MKK4/MKK7/JNK signaling pathway plays to control the positioning, morphology, and differentiation of the immature hippocampal subpopulation. This approach will make it possible to control adult pathways through the modulation of specific proteins that can be used as targets.

## Data Availability

Not applicable.

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
