# Peer review of "Dual Mkk4 and Mkk7 Gene Deletion in Adult Mouse Causes an Impairment of Hippocampal Immature Granule Cells"

_ijms, 2021, doi:10.3390/ijms22179545_

Round 1

Reviewer 1 Report

The authors of this study were the first to create conditional-induced KO mice with a genetic deletion for two genes Mkk4 and Mkk7. Received mice Actin-CreERT2 Mkk4flox / flox; Mkk7flox / flox made it possible to analyze the influence of these genes on neurogenesis in the hippocampus in adult animals. The work performed is a very modern and technically complex approach to the study of gene expression and the functional role of the proteins encoded by them. The work is relevant and the results presented in the manuscript seem to be important for understanding the regulation of the JNK pathway. However, there are multiple editorial problems, as well as several significant ones (listed below) that must either be eliminated or discussed and explained in the manuscript before publication. 

Abstract:

Lines 27-29: check the meaning of the sentence, please

Introduction:

Lines 53-55: check English and the meaning of the sentence, please

Lines 58-60: check English, please

Results:

Line 107: “in three different tissues of mice” - The figure shows the results for five different tissues.

Line 108: is showed – check English, please

Line 109: in graphs B for MKK4 and in graph C for MKK7 - Why do authors write “graphs” in the plural for B and the singular for C?

Line 109 and 110: Graphs bars -– check English, please

Line 118: “inbreed system” - check the spelling, please

Line 124: It is necessary to correct the order of description of the results presented in Figure 2. First, you should describe Figure 2a, then 2b, and only then 2c and 2d.

Lines 127-129: “ neither gene deletion nor protein elimination was observed in Actin-CreERT2

(Cre-/-) mice treated with tamoxifen, vehicle or in Actin-CreERT2 (Cre+/-) mice treated with

vehicle (Figure 2A-B).” - Probably, there should be a reference to Figure 2c as well.

Figure 2b: Figure 2b shows the deletion of only one allele, and the legends indicate that both alleles are deleted. Besides, the band for deletion is also seen in CRE(+)+V mice both for 7 days and 14 days. So, the text “This deletion is not observed in CRE(+)+V and in CRE(-)+T or CRE(-)+V groups.”(lines 137-138) is not correct for Mkk7 gene. The authors should explain Figure 2b, give the correct one or indicate that there was only a partial knockout for the gene and, accordingly, discuss this point.

Line 142: MKK$(a’,b’,c’) - check the spelling, please

Line 144: ; is needed at the end of the phrase “CRE(+)+V: CreERT2 (Cre+/-)- Mkk4 flox/flox,Mkk7 flox/flox plus vehicle”

 Line 159: probably A-D. Should be changed to A-E.

Line 167: “vehicle: Vehicle,” probably should be changed to “vehicle; V: Vehicle,”

Line 169: “levels in in different”

Line 176: “Vehicle,” probably should be changed to “V: Vehicle,”

Line 200: “The analyzes were” - check the spelling, please

 Lines 203-216: the text does not match what is shown in the linked figures. The text contains a link to figure 6d, but no figure.

Line 224: Please, define the DG.

 Line 229: “&&&p<0.001 vs WT” this was already indicated in line 228..

Lines 233-234: It is necessary to write more correctly what the authors meant in the phrase “cells ... were lower”

Line 239: co-loclization - check the spelling, please. No reference to Figure 8b. The phrase “co-localization was decreased” is not correct. Perhaps the authors wanted to write that the degree of co-loclization was decreased.

Line 242, 253, 258: “Mkk4 flox/flox Mkk7 flox/flox and Mkk4Δ/Δ,Mkk7Δ/Δ” - Throughout the text, you need to write the genotype using the same punctuation system

Line 244-245: «(c) I the right panel is shown in blue the nuclei by Hoeschst staining»- check the spelling and the meaning of the sentence, please

 Lines 247-248: Please check the correctness of the presence of this text in the captions to this figure.

Line 251 and 253:  mkk7 – correct, please

Lines 254-255: “Arrowheads indicate the same double-positive cell CR+ and DCX+ in channel-matched.” check the meaning of the sentence, please

Line 258: “A decrease is appreciated in Mkk4Δ/Δ,Mkk7Δ/Δ adult mice versus WT and Mkk4 flox/flox Mkk7 flox/flox” - Perhaps the authors wanted to write “the decrease is estimated”

Line 264: antibodies MAP2 are not described in Methods

Lines 266-268: check English

 Line 269: “stain revealed a lost of neural cells”-  check English

 Line 272: “Neuronal microtubule neurons are altered” - The phrase is written incorrectly, too general. It is necessary to indicate what exactly has been changed. And the meaning of the phrase also needs to be checked.

Line 276: check English

Line 280: In thepresent

Line 283-284: “The differential diminution evidenced that these proteins are higher in 283

CNS than in other tissues”.- In your experiment, this can already be seen from Figure 1. No knockouts are needed for this.

 Line 284: “In addition, the analyses showed higher levels of MKK4 than MKK7” - this is not shown in the manuscript.

Line 285: “The protein reduction”  - which of the two do you mean?

Line 291-293: judging by Figure 2b, your animals had only a partial knockout on the Mkk7 gene

Line 310: check the punctuation

 Line 311: Afte analyzing

Line 324: point needed

Line 329: check English

Line 335: punctuation; JIP proteins – define, please? And it is not clear why are ou talking about these protens

 Line 340; English

 Line 348: punctuation

 Line 358: please, Check if the “Mkk4flox or Mkk7flox” is spelled correctly

Line 365, 369, 374: punctuation

Methods:

4.1 Animals: It is not clear from the description of animals whether genes were completely removed or only their fragments, and if fragments, then which ones. The authors gave references to their previous works, but nevertheless, in this manuscript, a figure should be added that schematically explains the obtained genotypes of the experimental mice used in this work. In the same figure, you can indicate the location of primers for testing genotypes and deletions.

It is necessary to indicate on what number of animals each study was carried out and the repetition of the experiments.

Line 384 : matted  - probably the word mated should be used here

4.2 Tamoxifen treatment:  It is necessary to include in the text of the manuscript an image of the scheme of the experiment.

Line 411:  it is incorrect to write PCRs on tail, it is better to indicate that it was PCRs on tail DNA   

4.5 Immunoblot analysis: It is not described how the band size was determined on blots

Line 452: secondary antibodies are not described

 Line 486: why is the word gene used in the singular?

  1. Conclusion: The Conclusion reflects only the technical result. It is also necessary to describe the biologically important consequences of the resulting deletions.

Author Response

Dear reviewer,

We truly appreciate all the constructive comments and advices from the Reviewer 1. The followings are our point to point responses to the Reviewer’s comments. Our responses are highlighted with red color. Thank you for the opportunity to resubmit our work and for all the comments that will improve the manuscript.

We are sending to figures as supplements S1 and S2

Reviewer 2 Report

The manuscript studied the impact of Mkk4/Mkk7 dual deletion on adult mice. The dual deletion was carried out by a standard conditional-induced KO based on Actin-Cre system. The dual deletion is not completed – looks more like a KD. The authors validated the function of KD mostly at the protein level using IF, IHC and WB. They found the dual deletion in adulthood results in the alternation of dendritic patterns and differentiation of the neurons in the hippocampus through the phosphorylation of JNK. However, more molecular evidence is required to better support this finding – i.e. qPCR or RNAseq for critical JNK regulators.

Taken together, this is an incremental study with sufficient data. It is worthy to be published in IJMS after a minor revision. Please see my comments below:

  1. Please properly reference the important claims in section of result and discussion. For example, line 116, since Mkk4 and Mkk7 deletion results in a lethal embryonic phenotype. I noticed that the authors referenced the article in introduction. But it is better to cite them again for readability.
  2. The resolution of IF images is poor (i.e. Fig. 7 and 9). Please improve.
  3. The ethical statement regarding the animal work is missing.
  4. There are some typos in the current manuscript, i.e., line 290, ‘thepresent’ should be ‘the present’. Please correct.

Author Response

Dear reviewer,

We thank the reviewer 2 for the opportunity to resubmit our work and all the comments that will improve the manuscript. The followings are the responses in red color point to point to the Reviewer’s comments. We are adding Figure 6, 7 and 8 format tif

Best Regards

Carme Auladell and Ester Verdaguer

Round 2

Reviewer 1 Report

The authors have significantly improved the text of the manuscript, but the text still contains many inaccuracies and errors that must be corrected before publication. The list of errors I found is given below, however, this does not mean that I saw them all. Authors should carefully re-read the text on their own and check the presented data in order to exclude errors that may confuse the reader. In addition, authors should use the services of a professional translator to check the entire text, as the reviewer cannot be held responsible for pointing out all translation errors.

Line 24-25: “ To achieve this goal, Actin-CreERT2 24 Mkk4flox/flox,Mkk7flox/flox was created.” – probably, you have to add word “mice”.

Line 107: “of mice adulthood“ - English problems  

Figure 2. A. : Methods and Figure c1 show that the fragment for Mkk4 + (900 bp) and for Mkk4flox (1000 bp). Figure 2A shows that the fragment for Mkk4 + (850 bp) and for Mkk4flox (900 bp). An explanation and correction of this information is required.

Lines 124-126: English problems

 Line 148: “deletion of Mkk4 and Mkk7 gene is appreciated in “ - Perhaps the authors wanted to write “deletion of Mkk4 and Mkk7 gene is estimated”.

 Line 149-150: Immunoblot assay shows the progressive elimination 149 of MKK4 and MKK7 protein in the hippocampus, at 7 and 14 days after tamoxifen treatment in CRE(-) +T mice. - mice CRE (-) + T are specified incorrectly. Correction is needed.

 Line 155: “CRE(-)+V: Actin-CreERT2(Cre+/-)Mkk4flox/flox,Mkk7flox/flox plus vehicle;“ – please, make a correction according to Table 1

Lines 154-155: Actin-CreERT2 is used as just CreERT2 in definitions of CRE (+)+T and CRE(+)+V

Figure 4B: definition of constructions differs from those in Fig4A and Fig4C. Authors need to check and correctly label the tracks in the gel.

Line 177: “CRE(-)+T: Actin- CreERT2(Cre+/-)Mkk4flox/flox,Mkk7flox/flox plus tamoxifen, CRE(-)+V: Actin-CreERT2(Cre+/-)Mkk4flox/flox,Mkk7flox/flox “ – please, make a correction please, make a correction according to Table 1

Lines 176-177: Actin-CreERT2 is used as just CreERT2 for definition of several constructions

Lines 181-182: “WT, CRE(-)+V, CRE(-)+T  and CRE(-)+V” – do not corresponds to the definitions in Figure 4A-C

Line 203: Actin-CreERT2 is used as just CreERT2 for definition for one of constructions

Line 208: The analysis were performed - English problem

Line 226: “WT, Actin- CreERT2(Cre+/-) and Mkk4Δ/ΔMkk7Δ/Δ 226 mice”. This differs from the Figure and text above

Line 277: Hoeschst –spelling

 Line 285: extra point

Line 308: extra point

Line 334: “The levels of JNK phosphorylation decrease in Mkk4Δ/Δ,Mkk7Δ/Δ mice.“ -  English problems  

Line 346: “The induced deletion of JNK pathway alter” - English problem

Figure S1. Generation for Actin-CreERT2(+/-)Mkk4flox/flox ,Mkk7flox/flox .- You probably have to add word “mice”.                …. “which there is no recombinase”, …” which not express Cre recombinase “ – English problems

 Line 404 : protein. – should be in plural.

 Line 406: Mkk4flox/flox,Mkk7flox/flox. mice – extra point after flox

 Line 408: Mkk4Δ/ΔMkk7Δ/ Δ genotype – extra space in Δ/ Δ

 Line 445-446: English problems  

Figure S2.  “plus a pair of primer “ – it should beplus a pair of primers “.

Figure S2.  D. The right image of agarose gel for PCR deletion test: the size of band for Mkk7flox/flox is not defined in figure and not described in legend.

Line 472: “Size band was determianted” - English problems  and spelling problem

Author Response

Dear Reviewer,

Best Regards

Carme Auladell and Ester Verdaguer
